# Dataset on SCADA Data of an Urban Small Wind Turbine Operation in São Paulo, Brazil

Welson Bassi [1,2,*] , Alcantaro Lemes Rodrigues [1,2] and Ildo Luis Sauer [1,2]

1    Institute of Energy and Environment (IEE), University of São Paulo (USP), São Paulo 05508-010, Brazil
2    Center for Analysis, Planning and Energy Resources Development (CPLEN), São Paulo 05508-010, Brazil
*    Correspondence: welson@iee.usp.br

**Abstract:** Small wind turbines (SWTs) represent an opportunity to promote energy generation technologies from low-carbon renewable sources in cities. Tall buildings are inherently suitable for placing SWTs in urban environments. Thus, the Institute of Energy and Environment of the University of São Paulo (IEE-USP) has installed an SWT in an existing high-height High Voltage Laboratory building on its campus in São Paulo, Brazil. The dataset file contains data regarding the actual electrical and mechanical operational quantities and control parameters obtained and recorded by the internal inverter of a Skystream 3.7 SWT, with 1.8 kW rated power, from 2017 to 2022. The main electrical parameters are the generated energy, voltages, currents, and power frequency in the connection grid point. Rotation, referential wind speed, and temperatures measured in some points at the inverter and in the nacelle are also recorded. Several other parameters concerning the SWT inverter operation, including alarms and status codes, are also presented. This dataset can be helpful for reanalysis, to access information, such as capacity factor, and can also be used as overall input data of actual SWT operation quantities.

**Dataset:** https://doi.org/10.5281/zenodo.7348454

**Dataset License:** Creative Commons Attribution 4.0 International

**Keywords:** dataset; SCADA; operation; data; small wind turbine; urban wind energy; distributed generation; capacity factor; building adapted wind turbine; horizontal axis wind turbine (HAWT)

## 1. Summary

Several initiatives have been started and stepped up to reduce $CO_2$ emissions over the past few years, notably low-carbon energy generation technologies, particularly wind and solar. There has been a remarkable rise in the number of plants employing renewable energy and expansion of the distributed generation close to consumers, mainly thanks to photovoltaic systems but also due to the growing contribution of small wind turbines (SWTs) and other generating resources.

Tall buildings in urban areas are good locations for installing SWTs. Thus, the Institute of Energy and Environment of the University of São Paulo (IEE-USP) has installed a Skystream 3.7 SWT, with 1.8 kW rated power, in an existing high-height building on its campus in São Paulo, Brazil [1].

The installed SWT has the following specifications:

- Rated Capacity: 1.8 kW rated at 9 m/s, 2.4 kW peak at 13 m/s;
- Weight: 77 kg;
- Rotor Diameter: 3.72 m;
- Swept Area: 10.87 m$^2$;
- Type: Downwind rotor with stall regulation control;
- Alternator: Slotless permanent magnet brushless;

- Grid Feeding: 120/240 V ac / 59.3–60.5 Hz
- Yaw Control: Passive;
- Direction of Rotation: Clockwise looking upwind;
- Blade Material: Fiberglass-reinforced composite;
- Number of Blades: 3;
- Rated Speed: 50–330 rpm;
- Tip Speed: 9.7–63 m/s;
- Braking System: Electronic stall regulation with redundant relay switch control;
- Cut-in Wind Speed: 3 m/s;
- Cut-out Wind Speed: 25 m/s;
- Rated Wind Speed: 13 m/s;
- Survival Wind Speed: 226 km/h (63 m/s).

Figure 1 shows the geographical localization. The site's approximate GPS coordinates are −23.558160, −46.732895 (23°33′29.376″ S, 46°43′58.421″ W). The altitude of the terrain is around 740 m above sea level.

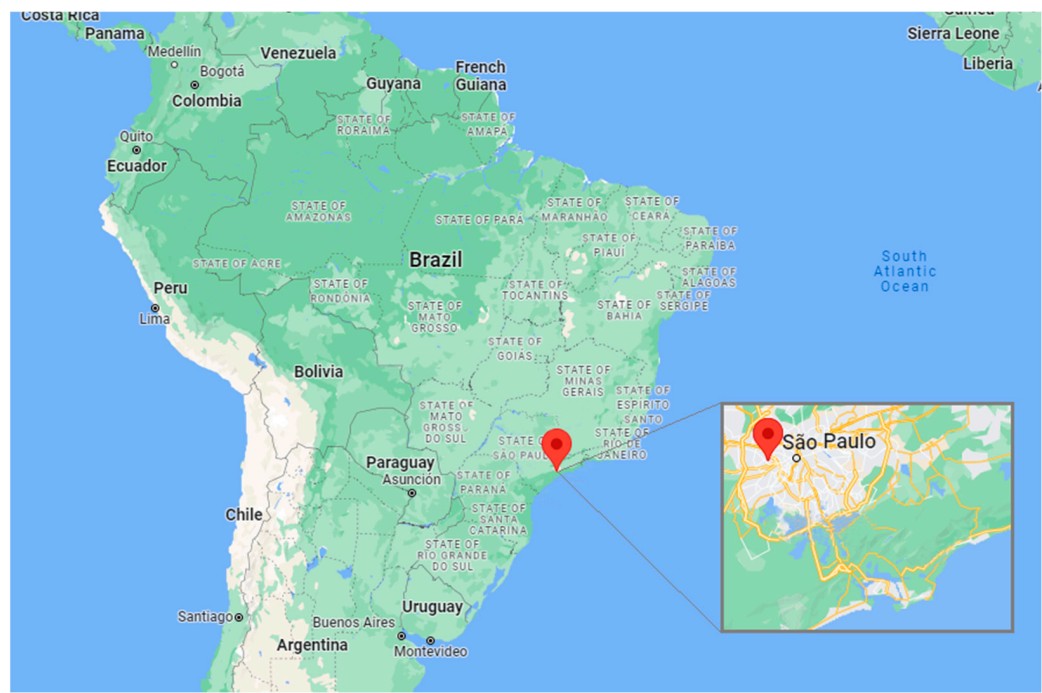

**Figure 1.** Geographical localization of the SWT.

Previous wind measurements with a remote sensing technology LIDAR (light detection and ranging) sensor, from December 2015 to January 2018, were recorded in a fixed location about 230 m away from the wind turbine installation point to assess the wind potential of the site. This remote sensor provides vertical height wind measurements, from 40 m to 290 m, stepped in 12 heights. The data relative to wind speeds at 40 m height (closest value of the installed height of the SWT hub) are summarized as follows: average wind speed 2.64 m/s, median wind speed 2.54 m/s, mean power density 19 W/m$^2$, and mean energy content 164 kWh/m$^2$/yr. Due to the relatively small wind speeds, a low-value capacity factor is expected. More detailed information is in [1].

The rooftop of the High Voltage Laboratory building at the Institute of Energy and Environment (IEE-USP) was selected to install the SWT due to the suitability of its location: namely, its height well above surrounding buildings and vegetation. Figure 2 shows implantation views.

SCADA data on wind turbines is not easily found in the literature. Usually, available datasets provide the geographic locations of wind turbines in operation in a particular country or region. An example of a dataset showing the location and technical description

of more than 70,000 wind turbines in the United States is in [2]. Another set that presents the spatial distribution, installed capacity, and commissioning year of wind turbines, photovoltaic systems, and bio and hydropower plants in Germany can be found in [3].

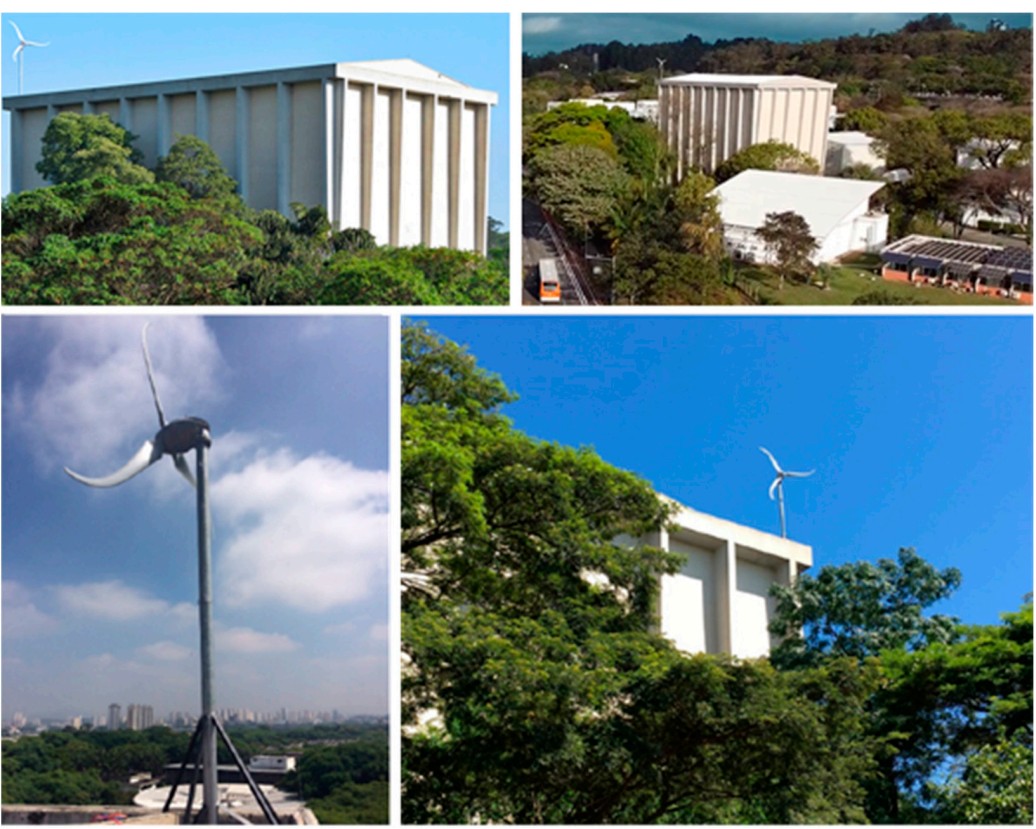

**Figure 2.** Views of the SWT implantation at the High Voltage Laboratory building of the University of São Paulo campus (São Paulo, Brazil).

In the literature, provided open access SCADA datasets could only be found for wind farms or big wind turbines [4–7], not covering information on small wind turbines.

During the IEE-USP SWT's operation, information from its SCADA data has been gathered, originating the current dataset. It contains data regarding the actual electrical and mechanical operational quantities and control parameters obtained and recorded by the internal inverter of the installed SWT during five full operation years, from 2017 to 2022. There were no mechanical or electrical failures, and no maintenance was required throughout this period.

This dataset can be helpful for reanalysis and can be used as input of actual SWT operation quantities.

## 2. Data Description

The current dataset is organized in per-year comma-separated values (.csv) files disaggregating the data along the years from 2017 (starting in November) to 2022, with one record per minute.

The files are named as "data_swt_iee_usp_YYYY.csv", where YYYY is the referring year. In the files, a semicolon symbol (;) is used as a column separator, while a dot symbol (.) represents the decimal separator. The first row of the CSV file corresponds to the header row to help identify data as described in the file "data_description.txt".

The first line on each year-based file represents the header table description of the data columns, as presented below. Table 1 shows the data description of the spreadsheets in the dataset.

**Table 1.** Data description of the spreadsheets in the dataset.

| Column | Record | Description |
|---|---|---|
| A | Log Time | Local time YYYY:MM:DD:HH:MM:SS.SSS |
| B | SN# | Serial number of the SWT. Value = 107605 |
| C | Software rev | The inverter firmware version. Value = 202 |
| D | opversion | The software version. Value = 300 |
| E | Inv Time | Decimal timestamp by the inverter (GMT) |
| F | watt-hours | Accumulated energy production (Wh) |
| G | Voltage In | Voltage generated by the rotor (V) |
| H | Voltage DC Bus | Voltage dc bus in the inverter (V) |
| I | Voltage L1 | Grid voltage, phase L1 (V) |
| J | Voltage L2 | Grid voltage, phase L2 (V) |
| K | voltage rise | Common voltage rise (V) |
| L | min v from rpm | Minimum voltage for the correspondent RPM (V) |
| M | Current out | Current ac (A) |
| N | Power out | Power output of the inverter (W) |
| O | Power reg | Power from generator (W) |
| P | Power max | Maximum power production of the inverter (W) |
| Q | Line Frequency | Frequency at the grid (Hz) |
| R | Inverter Frequency | Frequency at the inverter (Hz) |
| S | Line Resistance | Line impedance (Ohm) |
| T | RPM | Blade rotations per minute |
| U | Windspeed (ref) | Integer value of windspeed from RPM (m/s) |
| V | TargetTSR | Assigned tip speed ratio |
| W | Ramp RPM | Assigned rotational speed |
| X | Boost pulswidth | Pulse modulation at the inverter |
| Y | Max BPW | Maximum pulse modulation at the inverter |
| Z | current amplitude | Integer number proportional of current output |
| AA | T1 | Temperature at the heatsink T1 (Celsius) |
| AB | T2 | Temperature at the heatsink T2 (Celsius) |
| AC | T3 | Temperature at the nacelle (Celsius) |
| AD | Event count | Cumulative count of events |
| AE | Last event code | Code of the last event |
| AF | Event status | Code status of the current event |
| AG | Event value | Code value of the current event |
| AH | Turbine status | Current status code of turbine |
| AI | Grid status | Code status of the connected grid |
| AJ | System status | Code status of the system |
| AK | Slave Status | Code status of the system |
| AL | Access Status | Access level (signal strength) |
| AM | Timer | Timer countdown to zero after the last event |

As an illustration, Figure 3 shows an example of a single line taken from the dataset showing the above descriptors listed in Table 1 with their corresponding actual values. Due to the extensive length of characters (more than 400) and the columns (39), the example line is displayed in parts.

## 2.1. Data Analysis

Some data analysis derived from this dataset is provided in this section.

### 2.1.1. Power Quality Assessment

Some measured parameters make it possible to assess the power quality at the connection point with the low-voltage network. Indeed, a complete power quality analysis requires detailed and standardized measurements according to national regulations or international standards [8], but changes in voltage, current, or frequency are means of quantifying power quality. Therefore, an overview of power quality can be drawn by observing such variations over time.

| | A | B | C | D | E | F | G | H | I | J |
|---|---|---|---|---|---|---|---|---|---|---|
| | Log Time | SN# | Software rev | opversion | Inv Time | watt-hours | Voltage In | Voltage DC Bus | Voltage L1 | Voltage L2 |
| | 2022:01:08:14:23:01,103 | 107605 | 202 | 300 | 1641636050 | 1309353 | 253.1 | 375.7 | 132.8 | 136.8 |

| | K | L | M | N | O | P | Q | R | S | T |
|---|---|---|---|---|---|---|---|---|---|---|
| | voltage rise | min v from rpm | Current out | Power out | Power reg | Power max | Line Frequency | Inverter Frequency | Line Resistance | RPM |
| | 0 | 184 | 6.75 | 1529 | 1546 | 2400 | 59.97 | 59.88 | 252 | 315 |

| | U | V | W | X | Y | Z | AA | AB | AC | AD | AE |
|---|---|---|---|---|---|---|---|---|---|---|---|
| | Windspeed (ref) | TargetTSR | Ramp RPM | Boost pulswidth | Max BPW | current amplitude | T1 | T2 | T3 | Event count | Last event code |
| | 9 | 7 | 330 | 258 | 258 | 706 | 26.8 | 27 | 22.3 | 12518 | 111 |

| | AF | AG | AH | AI | AJ | AK | AL | AM |
|---|---|---|---|---|---|---|---|---|
| | Event status | Event value | Turbine status | Grid status | System status | Slave Status | Access Status | Timer |
| | 0 | 0 | 0 | 0 | 256 | 0 | 2 | 0 |

**Figure 3.** Example of a single line extracted from the dataset.

Figure 4 shows graphs related to voltage and frequency measurements provided by the SWT inverter. Ten-minute mean values and frequency histograms are presented for the five-year values contained in this dataset. The limits designated as 'Suitable Range', 'Precarious Range', and 'Critical Range' are defined according to the power quality regulation in Brazil, PRODIST Module 8 [9].

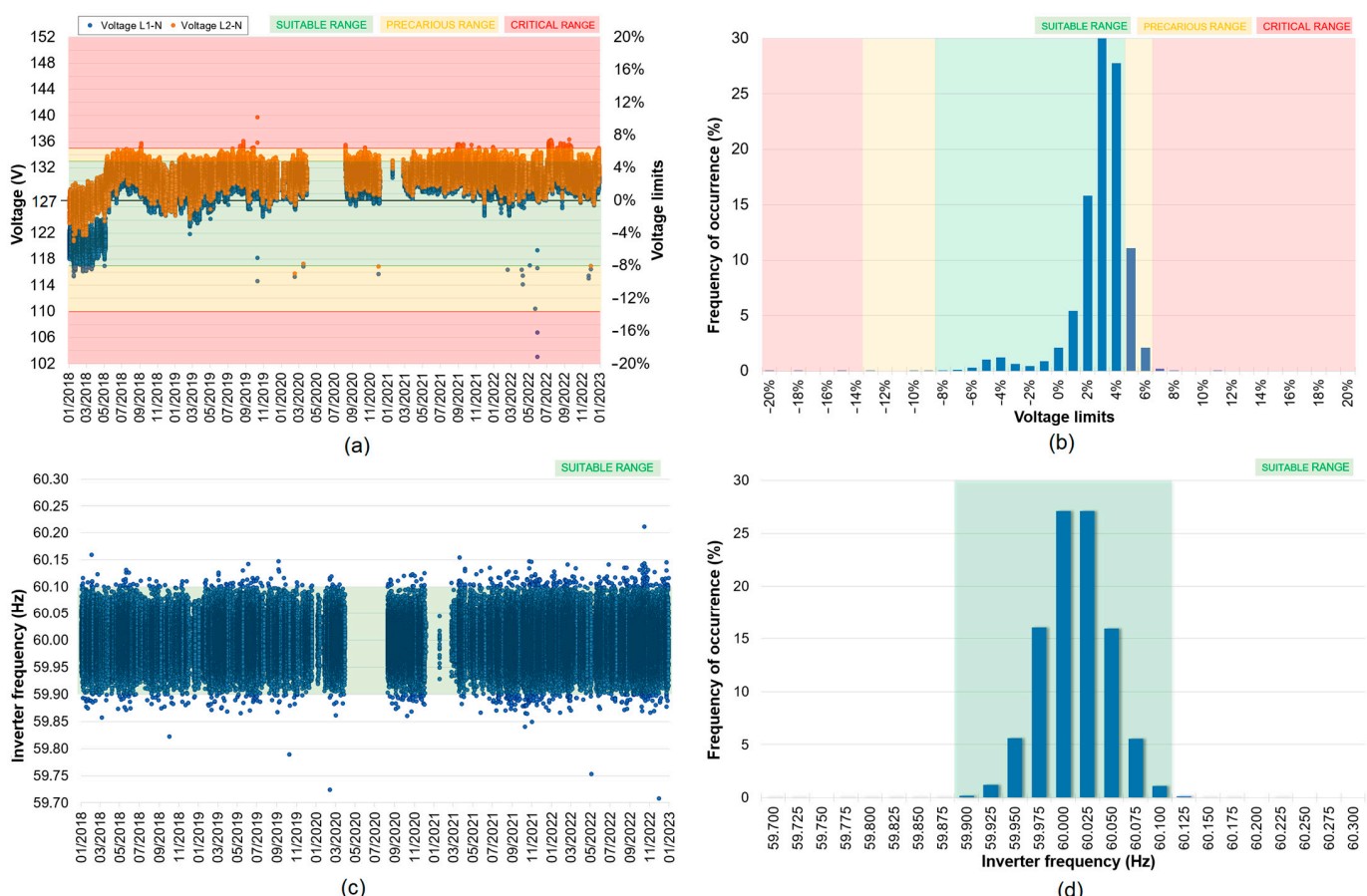

**Figure 4.** Ten-minute inverter electrical parameters. (**a**) Connection grid phase to neutral voltages recordings and (**b**) histogram of occurrences in the bin voltage limits. (**c**) Inverter frequency recordings and (**d**) histogram of occurrences in the bin frequency values.

The available measurements show that the PQ is good in terms of voltage and frequency, complying with the limits more than 99.8% of the time.

### 2.1.2. Energy Production

Figure 5 shows the energy production curve. The total produced energy in the five years was 1.6 MWh, leading to a capacity factor of about 2.0%.

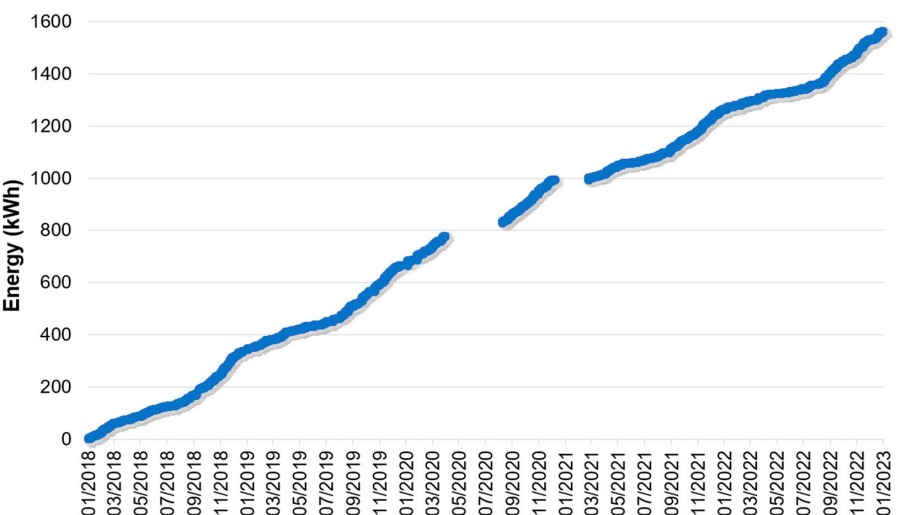

**Figure 5.** Accumulated produced energy from the whole years of 2018 to 2022.

### 2.1.3. Turbine Rotation and Wind Speed

Figure 6 shows 10-min mean values speeds in this dataset: the turbine rotation speed and the wind speed calculated by the inverter.

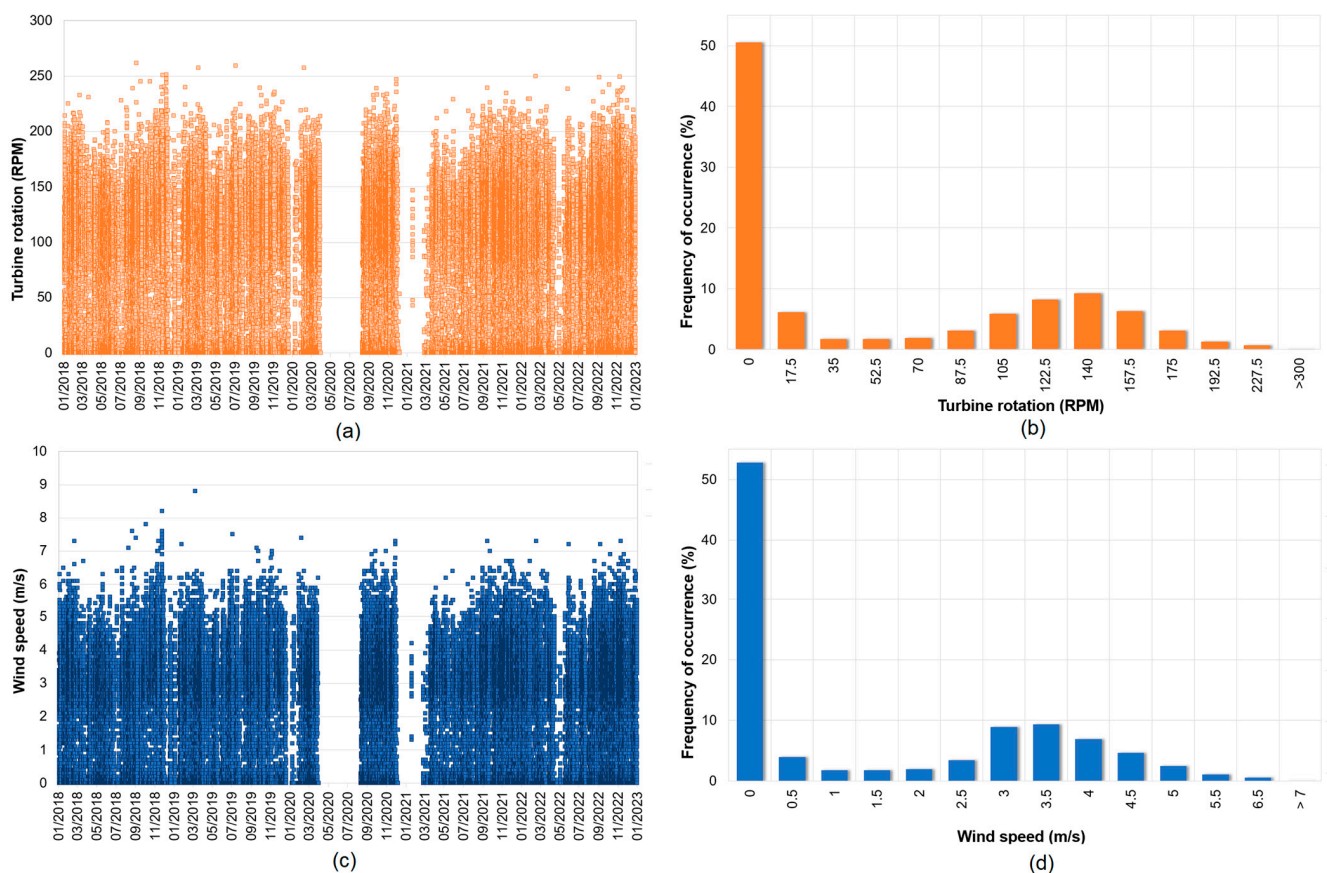

**Figure 6.** Ten-minute speed values. (**a**) Turbine rotation and (**b**) histogram of occurrences at the RPM bins. (**c**) Wind speed and (**d**) histogram of occurrences at wind speed bins.

Figure 7 shows 10-min mean boxplot diagrams and their yearly and monthly wind speed statistics.

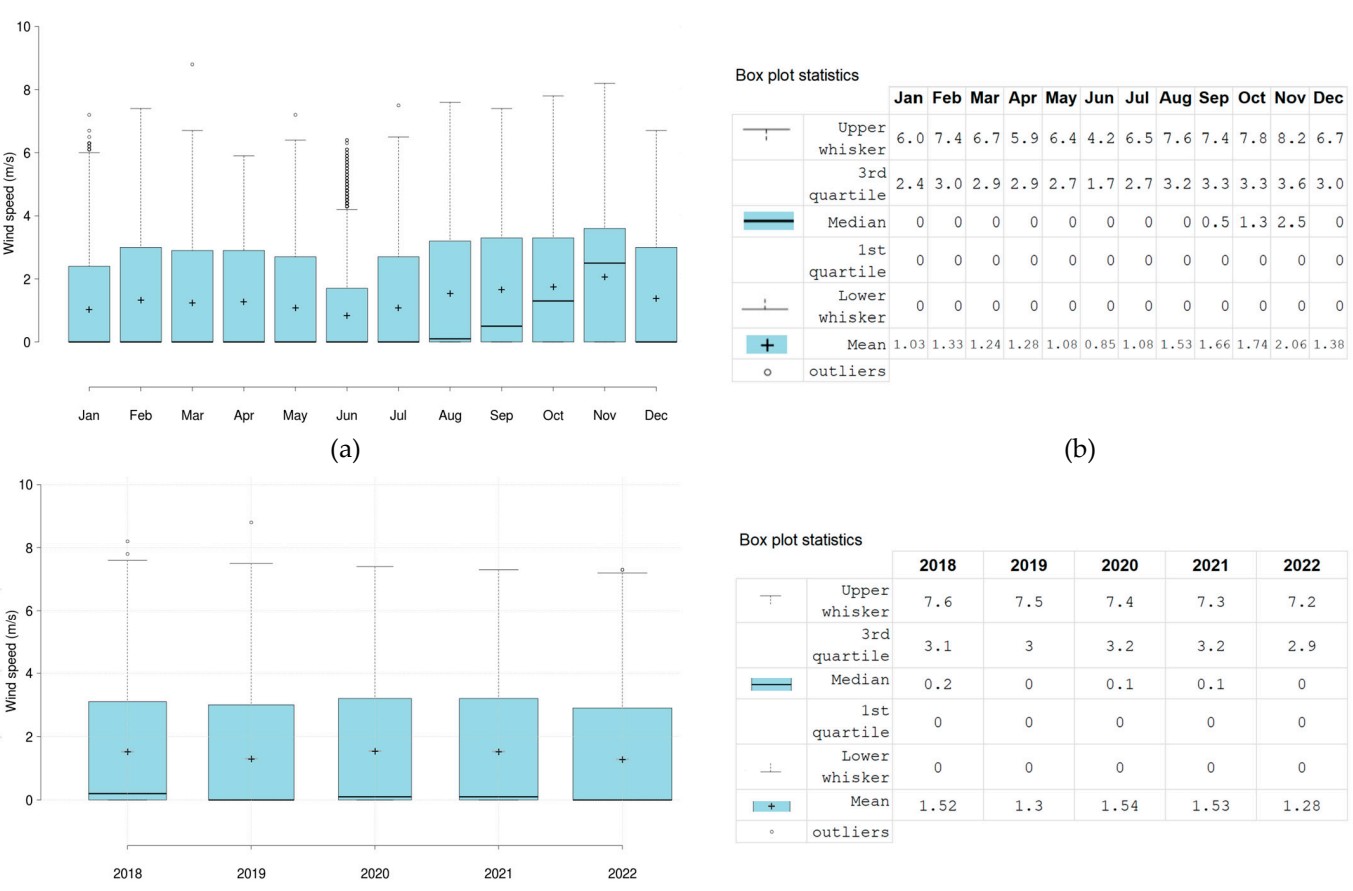

**Figure 7.** Ten-minute wind speed values calculations. (**a**) Monthly boxplot and (**b**) monthly boxplot statistics. (**c**) Yearly boxplot and (**d**) yearly boxplot statistics.

## 3. Materials and Methods

During the whole SWT operation, a supervising and monitoring facility setup was arranged to wireless receive, collect, and store the data from the wind turbine in order to analyze the operation.

The monitoring system comprises a Wireless PC Interface Adapter and Skyview 2.0.0 Software, both made by Southwest Windpower® and compatible with the Skystream 3.7 wind generator. The communication frequency is 2.4 GHz.

The Skyview 2.0.0 software runs on a Microsoft® Windows®-based microcomputer (PC), which connects with the internal inverter and controller of the small wind turbine (SWT) through an XBee® wireless protocol. The distance between the PC and SWT is about 100 m.

Figure 8 shows the Wireless PC Interface Adapter and some screen captures of the Skyview 2.0.0 Software.

The operational data are instantaneous and volatile information. The monitoring software was programmed to record all the available parameters.

The supervision system can store, among other quantities and parameters, electrical, mechanical, and operational information. The main electrical parameters are the energy, voltages, and currents in the connection grid point and power frequency. Mechanical information can be retrieved, such as the rotation and the referential wind speed. The temperature, measured in some points to the nacelle and inverter, is also recorded. Several other parameters concerning the SWT inverter operation, including alarms and status codes, are also presented.

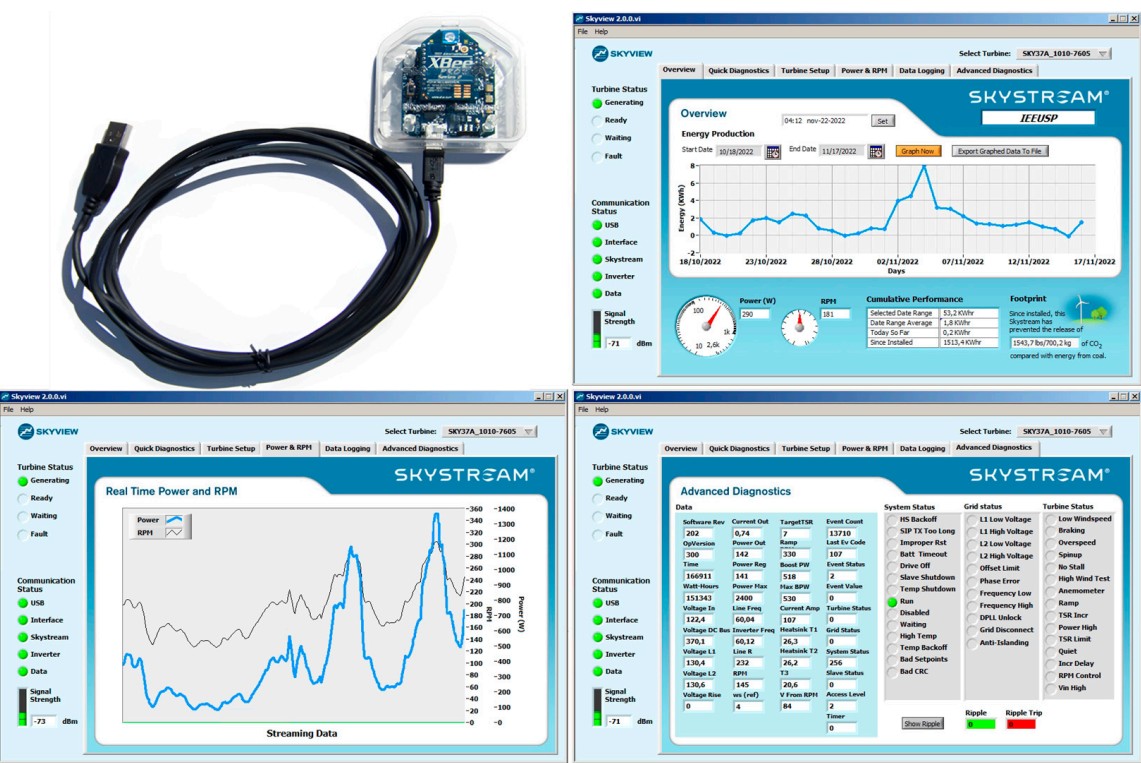

**Figure 8.** SWT's monitoring system components: wireless PC adapter 2.4 GHZ and screen captures of the monitoring software Skyview®.

## 4. Conclusions

This data descriptor article presents and describes a dataset containing data regarding the operation, from 2017 to 2022, of a small wind generator (SWT) (Skystream 3.7, 1.8 kW rated power), installed on an existing tall building at the University of Sao Paulo campus in the urban area of Sao Paulo, Brazil.

The data information was gathered by the SCADA system provided by the manufacturer in conjunction with a supervising and monitoring facility to wirelessly receive, collect, and store the data from the wind turbine´s inverter.

Some examples of information and analysis obtained directed from the dataset have been presented (e.g., power quality assessment on the connection point with the grid, rotation, and wind speed and energy production statistics).

This dataset can be helpful for reanalysis, as well as to access information about an SWT operation, and can be used as input data for SWT operation studies. It is not easily found in the literature SCADA data on wind turbine operations. When found, such datasets only usually provide data from wind farms or big wind turbines, not covering information on small wind turbines.

**Author Contributions:** Conceptualization: W.B., A.L.R. and I.L.S.; methodology: W.B., A.L.R. and I.L.S.; validation: W.B., A.L.R. and I.L.S.; formal analysis: W.B., A.L.R. and I.L.S.; investigation: W.B. and A.L.R.; resources: I.L.S.; data curation: W.B. and A.L.R.; writing—original draft preparation: W.B., A.L.R. and I.L.S.; writing—review and editing: W.B., A.L.R. and I.L.S.; visualization: W.B., A.L.R. and I.L.S.; supervision: W.B. and I.L.S.; project administration: W.B. and I.L.S.; funding acquisition: I.L.S. All authors have read and agreed to the published version of the manuscript.

**Funding:** The authors would like to acknowledge the financial support received from the Institute of Energy and Environment of University of São Paulo (IEE-USP) and from Enel Distribuição São Paulo in partnership with the Brazilian Electricity Regulatory Agency (ANEEL) through the Priority Energy and Strategic R&D entitled "Energy Efficiency and Minigeneration in Public University Institutions", grant number 00390-1086/2018.

**Data Availability Statement:** Data can be freely downloaded at https://doi.org/10.5281/zenodo.73 48454 (accessed on 20 January 2023) under Creative Commons Attribution 4.0 International License.

**Conflicts of Interest:** The authors declare no conflict of interest.

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
