# Peer review of "Dataset on SCADA Data of an Urban Small Wind Turbine Operation in São Paulo, Brazil"

_data, 2022_

Round 1

Reviewer 1 Report

In this paper, the authors shared a useful dataset of SCADA data from a small wind turbine. I have downloaded the dataset, and found that the dataset was available. The paper is well organized. The only comment is provided to the authors to improve the paper.

The SCADA dataset is complete. However, to make better use of the dataset, can you tell if the wind turbine has any major mechanical or electrical failure during the data collection process from 2017 to 2022? If possible, I look forward to receiving a corresponding descriptive document as a supplement to the dataset.

Author Response

Thanks, Review 1, for the comments.

During the operation period from 2017 to 2022, no mechanical or electric failures occurred, nor maintenance was required.

This sentence above was included in the manuscript.

Since no maintenance was required (as stated in Manufacturer Operation Manual), the authors do not have any maintenance record document to supplement it.

Reviewer 2 Report

The manuscript entitled "Dataset on SCADA Data of an Urban Small Wind Turbine Operation in Sao Paulo, Brazil" provides a comprehensive Dataset  for the Small Wind Turbines (SWT) applications. Important parameters, e.g. generated energy, voltages, currents, and power frequency, are included. The dataset is seen to be helpful for developing the actual SWT applications. Parameters for SWT, like SCADA, is rare in the literature. Thus it's important.

1. Table.1 should be reformatted. Sometimes it's capital; sometimes it's not.

2. For the outliers, as shown in Figure 3 and Fig. 6, is it needed to remove it? Can the author elucidate more on this?

Author Response

Thanks, Review 2, for the comments.

  1. The labels were placed in Table 1 precisely as received from the SCADA provided by the manufacturer: most of the labels in the header are in capital letters, but some others are not capitalized. So, to keep the exact correspondence between the data gathered from the SCADA and the published Dataset, the authors maintained the labels as presented in Table 1.
  2. Boxplot is a method of graphically representing groups of locality, dispersion, and skewness of numerical data by quartiles or a graphical representation for observation and further analysis. An outlier is an observation numerically distant from the rest of the data and is defined as a data point located outside the whiskers of the box plot. However, removing or ignoring outliers is generally not done because highlighting outliers is typically one of the advantages of using boxplots. The users and analysts can observe the overall descriptive data and make their decision on which extent of data is to be used. This Data Descriptor article is intended to illustrate just some possible utilization applications using the Dataset, not limited to the ones published here.

Reviewer 3 Report

The article describes a dataset recorded from a Small Wind Turbine installed on the rooftop of the Institute of Energy and Environment of the University of São Paulo, Brazil. The dataset contains electrical, mechanical quantities and control parameters recorded by the internal inverter of a Skystream 3.7, 1.8 kW wind turbine. The data are referred to the years 2017-2022. The main electrical parameters are the generated energy, voltages, currents, and power frequency in the connection grid point. Rotation, referential wind speed, and temperatures measured at the inverter and nacelle are also recorded. Among the possible applications, the authors state that this dataset can be helpful for reanalysis due to the lack of SWT public data or used as input data of actual SWT operation quantities. 

The research topic is relevant, but it should clarify the following points:

  • Provide an accurate description of the wind turbine, providing the technical specifications such as cut-in, rated and cut-off wind speed;
  • An overview of the wind potential of the site (urban contest) could help to understand the results and justify a low capacity factor;
  • After table 1, for better comprehension, you could add an example of a single line from the dataset describing each single component value;
  • About data quality: which quality control measures were adopted? What actions have been taken to reduce the influence of possible disturbances, such as noise? Please provide an accurate description.
  • Among the section of the paper, there are no discussions about the results or conclusions: detail the possible applications of the database, trying to describe the novelty compared to other data already available.

Author Response

Thanks, Review 3, for the comments.

  1. All the specifications were provided in Section 1
  2. The site's wind potential was published and analyzed in the authors' article in the Wind Journal (reference 1 of the paper). However, a summary of the wind resource measured with the Lidar sensor was also inserted in Section 1.
  3. A new Figure 3 was created illustrating a single line extracted from the Dataset.
  4. No effective energy quality was adopted since the inverter is certified, and the voltage fluctuations and frequency records show the limits are very adequate. These fluctuations are primarily imposed naturally by the existing grid, and the wind generator's presence does not contribute significantly to energy quality issues.
  5. According to the Data journal template for the authors, the Section Conclusion is optional. Considering the Reviewer 3 suggestion, a new section, Conclusions, was created to improve the article.